# The Rapid Data-Driven Prediction Method of Coupled Fluid–Thermal–Structure for Hypersonic Vehicles

**Jing Liu [1,2], Meng Wang [1] and Shu Li [1,\*]**

1   School of Aeronautic Science and Engineering, Beihang University (BUAA), Beijing 100191, China; bill16932@sina.com (J.L.); wangmeng0928@buaa.edu.cn (M.W.)
2   Science and Technology on Space Physics Laboratory, Beijing 100076, China
\*   Correspondence: lishu@buaa.edu.cn

**Abstract:** This work demonstrates the use of Latin Hypercube Sampling and Proper Orthogonal Decomposition in combination with a Radial Basis Function model to perform on vehicle prediction coupled fluid–thermal–structure. We regarded the Mach number, flight altitude and angle of attack as input parameters and established a rapid prediction model. The basic process of numerical simulation of the hypersonic vehicle coupled fluid–thermal–structure was studied to obtain the database of pressure coefficient, heat flux, structural temperature and structural stress as the sample data to train this prediction method. The prediction error was analyzed. The prediction results showed that the data-driven method proposed in this paper based on proper orthogonal decomposition and radial basis function could be used for predicting vehicle coupled fluid–thermal–structure with good efficiency.

**Keywords:** hypersonic vehicle; proper orthogonal decomposition; RBF model; data-driven; prediction



## 1. Introduction

A hypersonic vehicle is a challenging research area. This is due to the fact that researchers need to deal with a large flight envelope, rapidly changing aerodynamic coefficients, as well as coupling between the structure and the aerodynamics of the vehicle [1]. Design of structures [2] and thermal protection systems [3] for hypersonic vehicles depend on accurate predictions of the aerothermal loads, structural temperatures, and structural stresses [4–6]. Traditionally, a rigid isothermal body is assumed to analyze the surface pressures and heating rates [7]. These aerodynamic heating rates are used to analyze the structural temperature. The temperature and aerodynamic pressures are used to analyze the structural deformations and stresses [8,9]. The traditional independent approaches are inefficient because several iterations are required between the different analyses [10]. Aerodynamic heat is generated during the process of flight, which is transferred to the structure [11]. Thermal deformation of the structure is generated due to aerodynamic heat, which in turn affects the flow field. The problem should be simulated by the method of coupling fluid–thermal–structure. Pramote Dechaumphai et al. [12] carried out a study of a numerical simulation on leading edges coupled fluid–thermal–structure, and compared this with the experiment result. Many researchers such as Hirschel [2], Anderson [3], and so on had gone into studies on the aerodynamic heat of hypersonic vehicles. Apart from coupling, there is a large flight envelope. It is impossible to simulate the results of all flight conditions in the stage of engineering design [13,14], which will cost a lot of time. Therefore, it is necessary to develop a prediction method that is equivalent to CFD accuracy on vehicle coupled fluid–thermal–structure.

The Proper Orthogonal Decomposition (POD), also known as Principle Components Analysis (PCA), has been widely used for a broad range of applications. POD analysis yields a set of empirical modes, which describes the dominant behavior or dynamics of a given problem. This technique can be used for a variety of applications, including

derivation of reduced-order dynamical models, steady analysis and design of inviscid airfoils, image processing, and pattern recognition. Sirovich [15] introduced the method of snapshots as a way to efficiently determine the POD modes for large problems. In particular, the method of snapshots has been widely applied to computational fluid-dynamic (CFD) formulations to obtain reduced-order models for unsteady aerodynamic applications. A set of instantaneous flow solutions, or "snapshots" is obtained from a simulation of the CFD method. The POD process then computes a set of modes from these snapshots, which is optimal in the sense that, for any given basis size, the error between the original and reconstructed data is minimized. Reduced-order models can be derived by projecting the CFD model onto the reduced space spanned by the POD modes. Duan et al. [16] solved the inverse design problem of two-dimensional airfoil based on this method. Adding disturbance to the reference airfoil and performing CFD calculations to obtain sampling solutions. The airfoil inverse design can be performed based on the method of fitting the POD coefficients of the data, the airfoil shape corresponding to target pressure distribution can be achieved. Bui Thanh Tan [17] proposed to interpolate the POD coefficient under the condition of the known incoming flow state (such as Mach number, angle of attack) corresponding to the pressure coefficient distribution. This method can quickly obtain the approximate pressure coefficient distribution of the unknown flow field and greatly improves the calculation efficiency. The research object of these articles is very nearly flow field, and there are very few studies on the prediction of physical quantities coupled fluid–thermal–structure.

In this paper, we demonstrate the use of Latin Hypercube Sampling (LHS) and Proper Orthogonal Decomposition (POD) in combination with a Radial Basis Function (RBF) model to perform a prediction of the physical quantities coupled fluid–thermal–structure.

## 2. Rapid Data-Driven Prediction Method

In this section, we briefly review the theory of the POD and RBF Interpolation, followed by the rapid data-driven prediction method of a vehicle coupled fluid–thermal–structure, which are employed throughout this paper.

### 2.1. Proper Orthogonal Decomposition

The basic POD procedure is summarized briefly here. POD is based on the factorization of a spatial–temporal data matrix obtained from simulations or experiments [18]. The method is based on singular value decomposition (SVD) and attempts to find a linear lower-dimensional subspace than the original data. Through SVD, it is possible to obtain computed POD modes which are ranked in an energetic sense according to their residual variance [19].

Let $Y = [y_1, y_2, \ldots, y_n] \in \mathbb{R}^{p \times n}$ be the response matrix, which is obtained through experiments or simulations. In general, for numerical simulations, $p$ represents the number of model nodes, $n$ represents the number of snapshots. The purpose of POD is to find the optimal basis vector. The optimal POD basis vectors $u$ are chosen to maximize the cost:

$$\begin{cases} \max \frac{1}{n} \sum_{i=1}^{n} |(Y, u)|^2 \\ (u, u) = 1 \end{cases} \tag{1}$$

where $(Y, u)$ is the inner product of the basis vector $u$ with the field $Y$. Equation (1) can be solved by the Lagrange multiplier method:

$$J(u) = \sum_{i=1}^{n} \left( (Y_i, u)^2 - \lambda \left( \|u\|^2 - 1 \right) \right) \tag{2}$$

where $\lambda$ is the Lagrange multiplier, then take the derivative of $u$:

$$\frac{d}{du} J(u) = 2YY^T u - 2\lambda u \tag{3}$$

If we let $C = YY^T$, $C$ is a symmetric positive definite matrix. In general, for numerical simulations that generate big data problems, the snapshot POD method is more appropriate since it builds the covariance matrix with the $Y$ transposed data matrix. This leads to a covariance matrix that has a reduced size, but with the same eigenvalues and eigenvectors. Hence, the basis vectors are found by the eigen-decomposition problem:

$$Cu = \lambda u \tag{4}$$

where the eigenvalues $\lambda$ are positive and the eigenvectors $u$ form an orthonormal basis.

Consider energy, the larger the eigenvalue, the more energy it represents, and in terms of containing information, the larger the eigenvalue, the more information it has. The relative "energy" captured by the $i$th basis vector is given by:

$$I(r) = \frac{\sum_{i=1}^{r} \lambda_i}{\sum_{i=1}^{n} \lambda_i} \tag{5}$$

where $I(r)$ is energy ratio, the approximate prediction of the field $Y$ is then given by a linear combination of the eigenfunctions.

The basic POD procedure (just outlined) considers time-varying flows by taking a series of flow solutions at different instants in time [20]. In this paper, the procedure is applied in parameter space, that is, obtaining snapshots while allowing a parameter to vary. The parameter of interest could, for example, be the Mach number, angle of attack, and altitude.

### 2.2. Radial Basis Function Interpolation

Let $\hat{f}(x)$ be the original function to be modeled, and $f(x_i)$ be the values at $N$ discrete points $x_i$, $i = 1, 2, \ldots, n$, where $x_i$ is the vector of inputs at the $i$th sample point. The set of data points $X = \{x_1, x_2, \ldots, x_n\}$ is confined to a domain $\Omega$ in $n$-dimensional space, which is normalized to the unit hypercube $[0, 1]^n$ for interpolation [21].

The RBF model is a linear combination of basis functions, whose argument is the Euclidean distance between the point $x$ at which the interpolation is made and all the other points in the known dataset [22]. In other words, the interpolation at an untried site is a sum of contributions from all the known function values, the influence of which is controlled by a basis function that depends on the distance they are from the new site [23]. It provides a comprehensive and flexible interpolation method. Due to its good approximation, robustness and ease of implementation, the RBF model is used in many fields such as statistics, engineering optimization, and so on [24].

If $\varphi(x)$ is the chosen basis function and $\|\cdot\|$ is used to denote the Euclidean norm, the interpolation model has the form:

$$\hat{f}(x) = \sum_{i=1}^{n} \omega_i \cdot \varphi(\|x - x_i\|) \tag{6}$$

where $\omega_i$, $i = 1, 2, \ldots, n$ is the weight coefficient. The coefficients are found by requiring exact recovery of the original data $\hat{f}(x) = f(x)$ for all points in the training dataset $X$.

When there are no repeated sample points, the radial function $\varphi(x)$ is positive The interpolation coefficients $\lambda_1, \lambda_1, \ldots, \lambda_n$ can be found by solving the linear system problem:

$$\underbrace{\begin{bmatrix} \varphi(\|x_1 - x_1\|) & \varphi(\|x_1 - x_2\|) & \cdots & \varphi(\|x_1 - x_n\|) \\ \varphi(\|x_2 - x_1\|) & \varphi(\|x_2 - x_2\|) & \cdots & \varphi(\|x_2 - x_n\|) \\ \vdots & \vdots & \ddots & \vdots \\ \varphi(\|x_n - x_1\|) & \varphi(\|x_n - x_2\|) & \cdots & \varphi(\|x_n - x_n\|) \end{bmatrix}}_{A} \begin{bmatrix} \omega_1 \\ \omega_2 \\ \vdots \\ \omega_n \end{bmatrix} = \begin{bmatrix} f(x_1) \\ f(x_2) \\ \vdots \\ f(x_n) \end{bmatrix} \tag{7}$$

where $A$ is the interpolation matrix. The obtained RBF interpolant $\hat{f}(x)$ can be used to approximate the given function $f(x)$.

### 2.3. Rapid Prediction Method Based on Data-Driven

In our work, we made predictions on the hypersonic vehicle coupled fluid–thermal–structure, which concerned heat flux of flow field, pressure coefficient of the flow field, structural temperature and structural stress. POD provides a method of processing the dataset [25–28]. Thereby the specific process of the prediction method [29–32] is introduced in detail next, and the corresponding flow chart is shown in Figure 1.

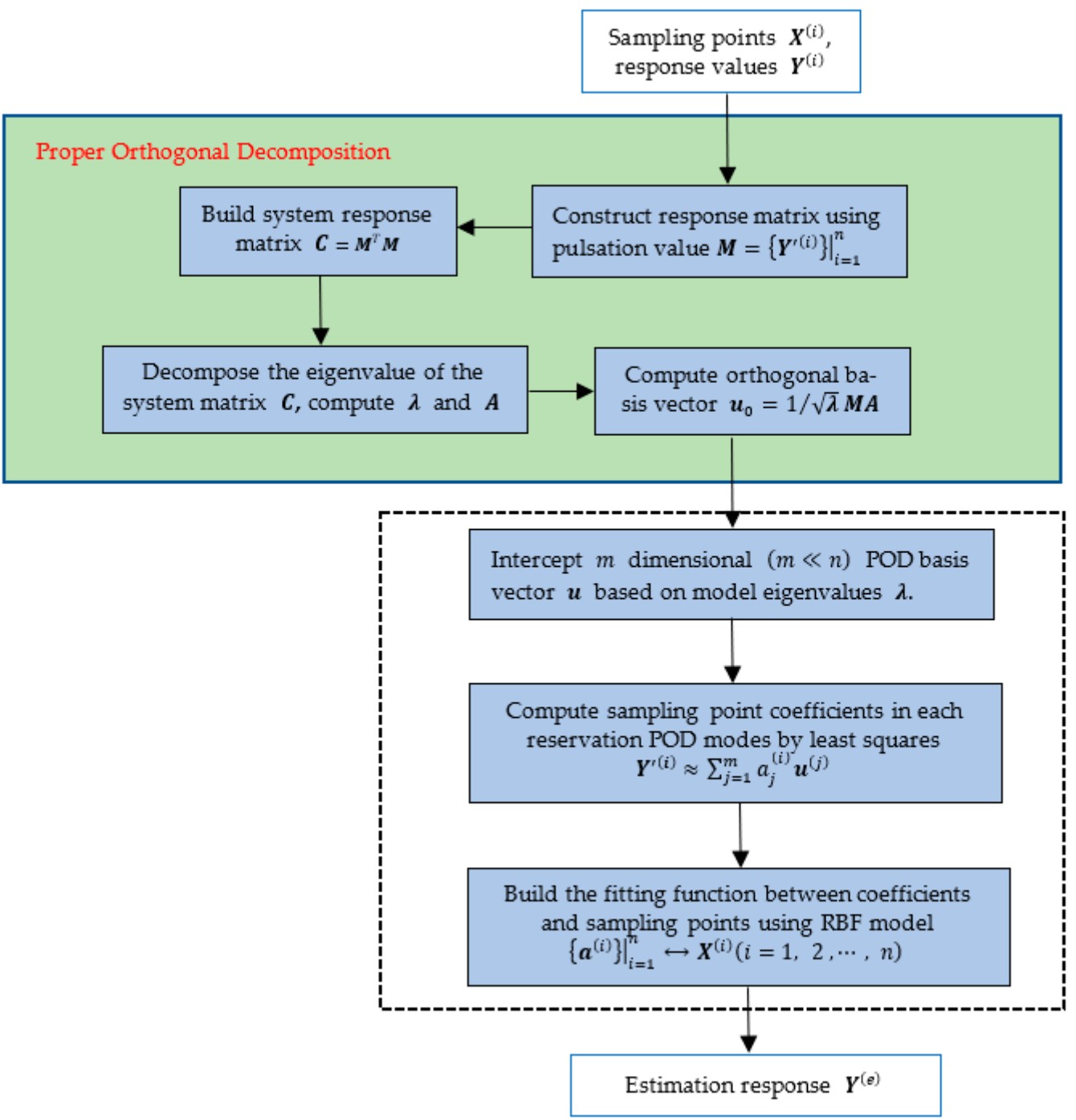

**Figure 1.** Flowchart of prediction of the data-driven method.

Firstly, the state variables and their range is selected according to the flight status. Latin Hypercube Sampling is used to obtain sample points *X* in the design space as shown in Figure 2. The state variables and their range is shown in Table 1.

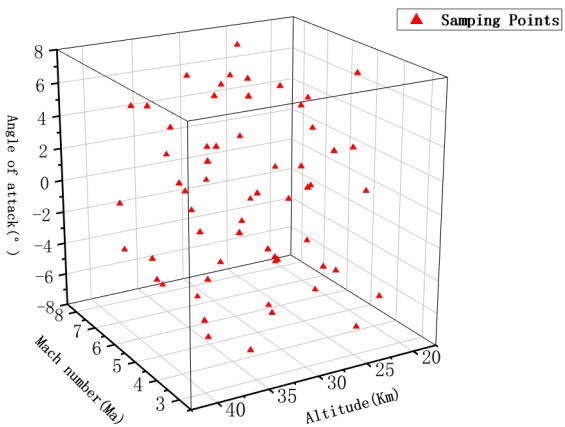

**Figure 2.** Sampling points.

**Table 1.** State variables and bounds of variables.

| Mach Number (Ma) | Altitude (H/km) | Angle of Attack (α/°) |
| :---: | :---: | :---: |
| $3 \leq Ma \leq 8$ | $20 \leq H \leq 40$ | $-8 \leq \alpha \leq 8$ |

Secondly, numerical simulation is adopted to obtain the response matrix of all sample points $Y$. The response matrix $Y$ can be expressed as the superposition of the average value $\overline{Y}$ and the pulsation value $Y'$, that is $Y = \overline{Y} + Y'$. The pulsation value $Y'$ is used to construct matrix $M$:

$$M = \begin{bmatrix} Y_1'^{(1)} & Y_1'^{(2)} & \cdots & Y_1'^{(n)} \\ Y_2'^{(1)} & Y_2'^{(2)} & \cdots & Y_2'^{(n)} \\ \vdots & \vdots & & \vdots \\ Y_p'^{(1)} & Y_p'^{(2)} & \cdots & Y_p'^{(n)} \end{bmatrix} \tag{8}$$

where $n$ is the number of sampling points; $p$ is the number of model nodes; $Y'$ is pulsation response value of model nodes, which could be heat flux, pressure coefficient, structural temperature and structural stress in this paper.

Thirdly, pulsation response matrix $M$ is used to construct the correlation matrix $C$. Then, we decompose the eigenvalue of $C$, $C = M^T M$, $CA = \lambda A$, $u_0 = 1/\sqrt{\lambda}MA$, where $\lambda$ and $A$ respectively correspond to eigenvalue and eigenvector, $u_0$ is the orthogonal basis vector.

Fourthly, eigenvalues are sorted from large to small, and the larger the eigenvalue, the more energy it represents. POD basis vector $u$, which is $m$ dimensional $(m \ll n)$, is selected according to $\lambda$.

Fifthly, the POD basis vector is obtained in the fourth step. From the approximate relationship $Y'^{(i)} \approx \sum\limits_{j=1}^{m} a_j^{(i)} u^{(j)}$, the coefficients $a$ of the POD basis vector under all sample points can be calculated by the least square method. Due to the relationship between sample points $X$ and coefficients $a$, RBF is used to establish their approximate fitting relationship [33–38].

Sixthly, the argument of RBF is the Euclidean distance between the point $x$ at which the interpolation is made and all the other points in the known dataset. Given a test sample point $X^{(e)}$ in the design space, the corresponding coefficient $a^{(e)}$ can be obtained, followed by response value of physical quantities can be obtained according to $Y^{(e)} = \overline{Y}^{(i)} + \sum\limits_{j=1}^{m} a_j^{(e)} u^{(j)}$.

### 3. Numerical Simulation

The numerical simulation coupled fluid–thermal–structure was studied to obtain the response value of sampling points to establish a database set. The simulation should be verified to ensure that the database we build can accurately contain the information of the flow field and structure.

The equations for unsteady compressible flow are described by the conservation of mass, momentum, and energy equations [12]. These equations can be written in the conservation form as:

$$\frac{\partial}{\partial t}\{Q_F\} + \frac{\partial}{\partial x}\left\{E_F^I + E_F^V\right\} + \frac{\partial}{\partial y}\left\{G_F^I + G_F^V\right\} = 0 \tag{9}$$

where $\{Q_F\}$ is the vector of the conservation variables; $\{E_F^I\}$ and $\{G_F^I\}$ are vectors of the inviscid flux components in the $x$ and $y$ directions; $\{E_F^V\}$ and $\{G_F^V\}$ are vectors of the viscous flux components in the $x$ and $y$ directions.

For transient heat conduction without an internal heat source, the thermal response of the structure is described by the energy equation that can be written in conservation form as:

$$\frac{\partial}{\partial t}(Q_T) + \frac{\partial}{\partial x}(E_T) + \frac{\partial}{\partial y}(F_T) = 0 \tag{10}$$

where $Q_T$ is the conservation variable, $E_T$ and $F_T$ are the component of flux.

The structural response is described by the quasistatic equilibrium equations that can be written in conservation form as:

$$\frac{\partial}{\partial t}(Q_s) + \frac{\partial}{\partial x}(E_s) + \frac{\partial}{\partial y}(F_s) = 0 \tag{11}$$

where $Q_s$ is the displacement vector; $E_s$ and $F_s$ are vectors of the stress components.

An experiment was performed in the NASA Langley 8-ft High-Temperature Tunnel in 1987. The NASA Langley Research Center 8-ft High-Temperature Tunnel (8-ft HTT) is a hypersonic blowdown tunnel in which a high energy level for simulating hypersonic flight is obtained by burning methane and air in a high-pressure combustor. A 3-inch diameter and 0.5-inch thick, stainless-steel cylinder was mounted on the panel holder and subjected to a uniform high-enthalpy Mach 6.47 flow. The solution to Mach 6.47 flow over a cylinder is used to demonstrate the integrated fluid–thermal–structure analysis approach. The schematic diagram of this 8-ft HTT can be found in Figures 8 and 9 of [10]. And more details of the experimental configurations, the tunnel flow conditions, and the experimental results can be found in [10].

Because of symmetry, the model is reduced to two dimensions. Only one-half of the incoming flow domain and cylinder is modeled. In this paper, the finite-element model representing the flow domain and the cylinder is shown in Figure 3. The number of grids of fluid is $150 \times 400$. The number of grids of structure is $100 \times 13$. The grid near the wall of fluid is encrypted, and the minimum height is $10^{-6}$ m. In order to better capture the characteristics of the flow field, the grid at the location of the shock wave is encrypted.

The hypersonic aerodynamic heat is calculated by FLUENT and the structural heat transfer is calculated by ABAQUS. The pressure far field boundary condition is used in the flow domain and the $SST\ k-\omega$ turbulence model is used in the whole flow computational domain. A no-slip condition is specified at the surface of the wall. The analysis step of coupling temperature-displacement is used in the cylinder and the initial temperature of the cylinder is 294.4 K. The code is used to realize data exchange in the process of simulation through the surface of the wall between ABAQUS and FLUENT. The incoming condition is shown in Table A1. The material property is shown in Table A2.

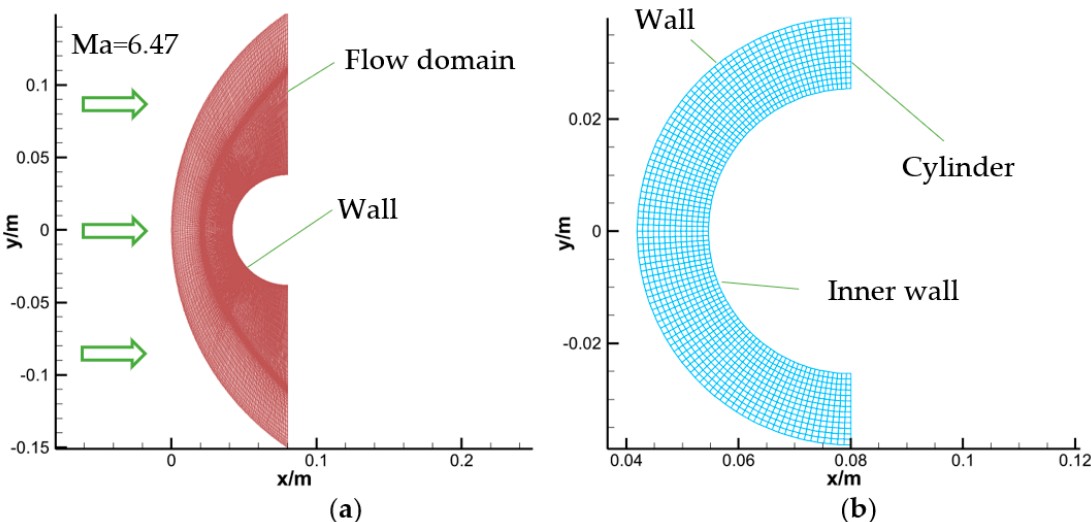

**Figure 3.** Finite-element model: (**a**) flow domain; (**b**) cylinder.

The result of the numerical simulation is shown in Figure 4 and Table 2. Density contours are compared in Figure 4 with an experiment result [10]. The comparison of the shock shape and position indicates the global flow field is reasonably well obtained. We also compared the result of the maximum temperature of structure, maximum heat flux and the maximum pressure of flow field between the numerical simulation in this paper and experiment, which is shown in Table 2. We could conclude that the numerical simulation used in this paper could obtain the information of flow field and structure for establishing a database. For the experimental results and how they were obtained please refer to [10].

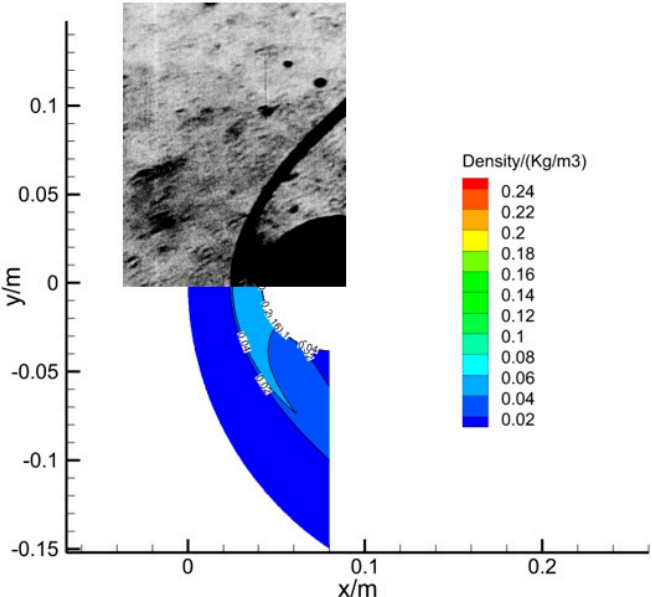

**Figure 4.** Comparison of density contours.

**Table 2.** Comparison of numerical simulation in this paper with the experiment.

| Method | Maximum Temperature of Structure/K | Maximum Heat Flux/(kW/m$^2$) | Maximum Pressure of Flow Field/Pa |
|---|---|---|---|
| Numerical Simulation in this paper | 436 | 663 | 35,386 |
| Experiment [10] | 465 | 670 | 37,815 |

## 4. Result of Prediction

Five test conditions were selected according to the Latin Hypercube Sampling method. We take the arithmetic average of these predicted results. The error analysis is performed on an arithmetic average. The test conditions are shown in Table 3.

**Table 3.** Test sampling point.

| Test Condition | Altitude (H/km) | Mach Number (Ma) | Angle of Attack ($\alpha$/°) |
|---|---|---|---|
| 1 | 38 | 4.2 | −3.2 |
| 2 | 26 | 5 | −6.4 |
| 3 | 30 | 3.4 | 3.2 |
| 4 | 34 | 6.6 | 0 |
| 5 | 22 | 5.8 | 6.4 |

Heat flux and pressure coefficient of the wall of the fluid domain are usually of interest to researchers. By comparing the actual value with predicted value, the accuracy of the prediction method could be judged. The wall consists of 355 nodes, the actual pressure coefficient and predicted pressure coefficient of these nodes are shown in Figure 5. The actual heat flux and predicted heat flux of these nodes are shown in Figure 6. It could be seen that the predicted value is almost consistent with the actual value regardless of the pressure coefficient or the heat flux.

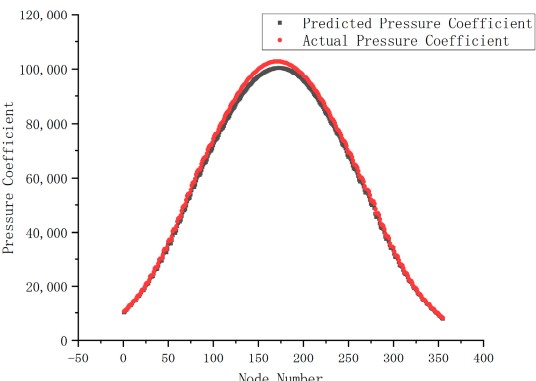

**Figure 5.** Predicted and actual pressure coefficients.

The contours of actual and predicted results, including temperature and stress, are shown in Figures 7 and 8. Through comparing actual and predicted results, it could be seen that the actual result is similar to the predicted result in terms of value and distribution.

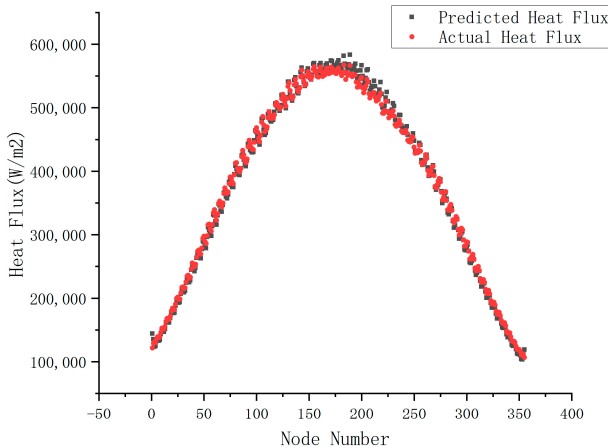

**Figure 6.** Predicted and actual heat fluxes.

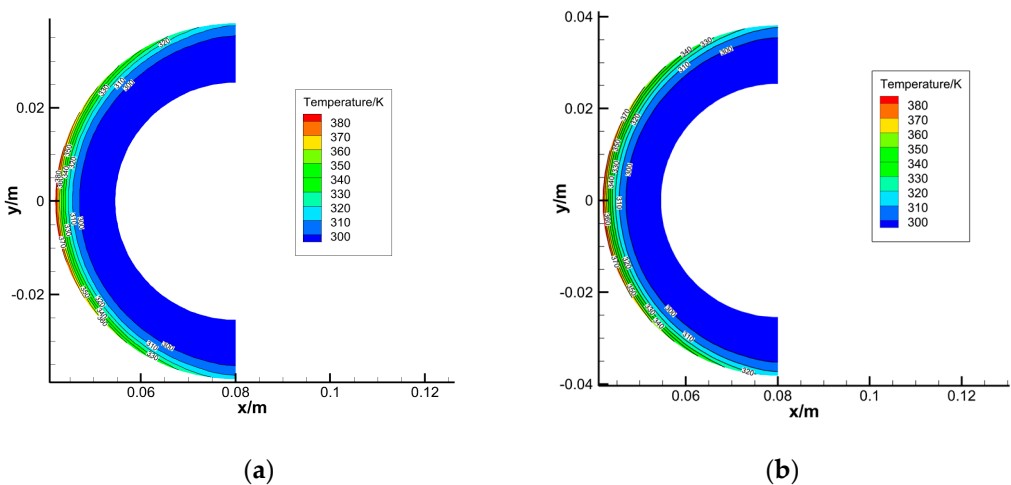

(**a**)                                            (**b**)

**Figure 7.** The contour of structural temperature: (**a**) actual temperature; (**b**) predicted temperature.

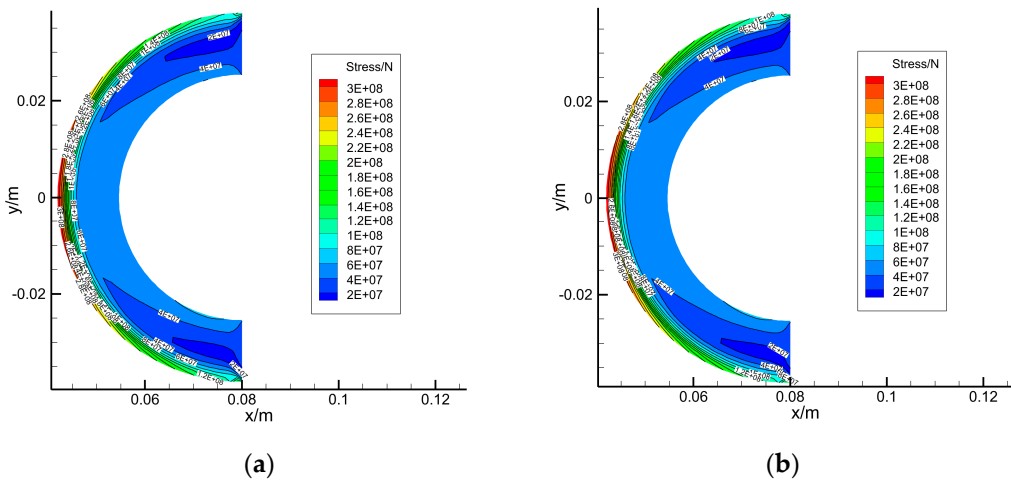

(**a**)                                            (**b**)

**Figure 8.** The contour of structural stress: (**a**) actual stress; (**b**) predicted stress.

In order to make the results more intuitive, the relative error of the structural temperature and stress are shown in Figure 9.

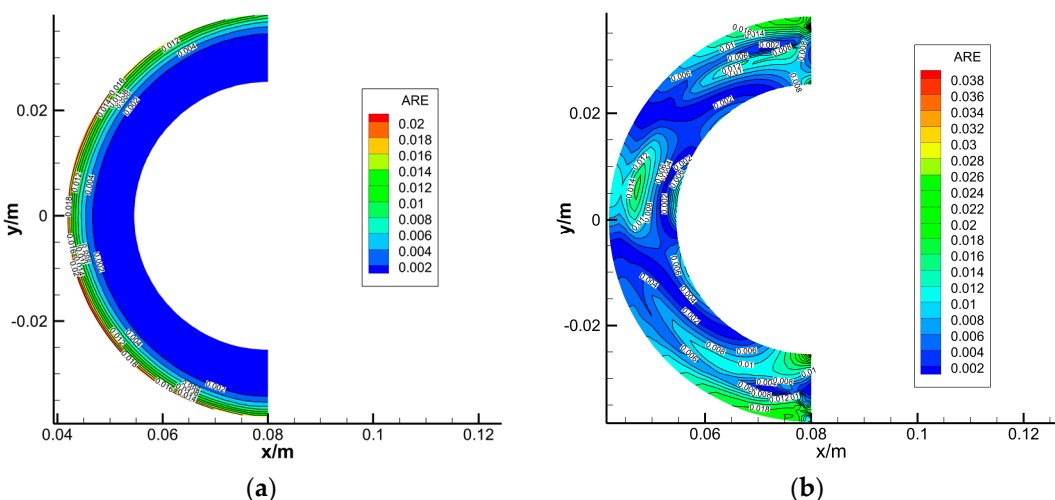

**Figure 9.** Relative error contour of the structural temperature and stress: (**a**) temperature; (**b**) stress.

It could be seen that the maximum relative error of structural temperature is only 2%, which is practically located at the front of the structure. The maximum relative error of structural stress is only 3.8%, which appears in very few nodes. However, the test sets of data were simply to verify the prediction effect of the method. There might have been a few errors between the different sets of data that could be reduced by obtaining more sets of verification data. The prediction error can be reduced by increasing the sample data.

The comparison of efficiency between the predicted method and numerical simulation is shown in Table 4. The computer is configured as AMD Ryzen 7 4800 H, 2.9 GHz, 16.0 GB RAM. The calculation with the software took nearly 12 h. It is difficult to obtain the information of flow field and structure quickly. However, it only took 0.14 s to complete the calculation by using the prediction method proposed in this paper.

**Table 4.** Comparision of efficiency with prediction method and numerical simulation.

| Method | Number of Snapshots | CPU Time for One Snapshot/h | CPU Time for Predicting one Test Condition/s |
|---|---|---|---|
| Prediction method | 5 | ≈12 | 0.14 |
| Numerical simulation | 60 | ≈12 | – |

## 5. Conclusions

In this paper, the prediction accuracy of the data-driven method on vehicle coupled fluid–thermal–structure was investigated for pressure coefficient, heat flux, structural temperature and structural stress by use of the POD and RBF models. According to the analysis of Section 4, it could be seen that the data-driven method proposed in this paper could be used for predicting the physical quantity of a hypersonic vehicle coupled fluid–thermal–structure with good efficiency. Compared with the numerical simulation method, the data-driven prediction method had the advantages of low cost and high speed. If sufficient data can be provided, the calculation result will be consistent with the actual values.

In future work, in order to improve the accuracy of the prediction method, we will study the relationship between the accuracy of the prediction result and the size of the database. The combination of proper orthogonal decomposition with other machine learning models will be studied, and we hope to improve the generalization ability of the prediction methods.

**Author Contributions:** Conceptualization, J.L., M.W. and S.L.; methodology, J.L. and M.W.; formal analysis, J.L.; writing—original draft preparation, J.L.; writing—review and editing, J.L. and M.W. All authors have read and agreed to the published version of the manuscript.

**Funding:** This research received no external funding.

**Institutional Review Board Statement:** Not applicable.

**Informed Consent Statement:** Not applicable.

**Data Availability Statement:** Not applicable.

**Conflicts of Interest:** The authors declare no conflict of interest.

### Appendix A

The incoming flow conditions and material properties of the stainless-steel cylinder are given in the following table:

**Table A1.** The incoming flow conditions.

| $P_\infty$(Pa) | $T_\infty$(K) | *Ma* | $Re_\infty$ |
|---|---|---|---|
| 648.1 | 241.5 | 6.47 | $1.31 \times 10^6$ |

**Table A2.** The material property of the stainless-steel cylinder.

| Elastic Modulus (GPa) | Poisson's Ratio | Density (Kg/m$^3$) | Coefficient of Linear Expansion $\times 10^{-6}$(K$^{-1}$) | Thermal Conductivity (W/(m·K)) | Specific Heat Capacity (J/(kg·K)) |
|---|---|---|---|---|---|
| 206 | 0.3 | 8030 | 17.5 | 16.27 | 502.48 |

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
