# Peer review of "The Rapid Data-Driven Prediction Method of Coupled Fluid–Thermal–Structure for Hypersonic Vehicles"

_aerospace, doi:10.3390/aerospace8090265_

Round 1

Reviewer 1 Report

This article presents a method for fast prediction of hydrodynamic parameters for a vehicle such as pressure coefficient, heat flux, structure temperature, and structural stress was investigated using Proper Orthogonal Decomposition and Radial Basis Function models. The data management method proposed in this article can be used to predict the physical quantities of a hypersonic vehicle with a liquid-thermal structure with good efficiency. Comparison of the data obtained using the prediction method with the calculated data in the simulation of hypersonic flow showed close agreement. But compared to numerical simulation, the data-driven prediction method had the advantages of low cost and high speed. The given prediction method can be used in the development and design of hypersonic vehicle. The work is relevant and contains an original prediction method. The article can be published.

Author Response

Please see the enclosed.

Reviewer 2 Report

The work is good, but there are comments.

The title must contain hyphens, not dashes. Not "...Fluid-Thermal –Structure...", but "...Fluid-Thermal-Structure...".

Abstract: "In our work, we demonstrated...". Maybe "The work demonstrates..." would be better?

There are many good works and review papers on hypersonic aerothermodynamics. Why aren't they referenced in the introduction? For example Ya. B. Zel’dovich, Yu. P. Raizer. Physics of Shock Waves and High-Temperature Hydrodynamic Phenomena; J D Anderson. Hypersonic and High-Temperature Gas Dynamics; EH Hirschel. Basics of aerothermodynamics; JS Shang, ST Surzhikov. Plasma Dynamics for Aerospace Engineering; and so on.

In the introduction, I don't understand the meaning of references to sources [1-3]. You talk about hypersonics, aerodynamic coefficients, aerodynamic heat affection on flow field,  simulation of flight conditions. What does it have to do with Sandwich Plates, Supersonic Nozzles, Bionic Surfaces, Transition Pieces?

Maybe make Fig. 1 sharper and clearer? And add "°" and "km" at AoA and Altitude.

In general, pay attention to all the figures and captions. Somewhere the scale has moved out (fig.8a), somewhere the actual values are not visible (figs.8,10a). Don't be lazy, make your article neat and enjoyable to look at and read.

line 173: Why [37]? There are not even such words about unsteady compressible flow in that work.

Where did experiment of Mach 6.47 come from? In [25], [11], I did not find it. Maybe pages were missed. In any case, you need to clarify this and add the correct source. And add a link to the caption of Figure 3. And change the text on line 193 to something like "Experiment diagram from [...] for coupled fluid‐thermal‐structure analysis".

Where are boundary and initial conditions of performed numerical simulation? At least a brief description should be presented.

line 201: "The material Property is shown in Table A2"

Change the text on line 204 to "Finite‐element model of flow domain (a) and cylinder (b)"? Same goes for captions of Figs. 9,10.

Why is the thickening of the mesh in the region of the shock wave not described? Otherwise, the reader will have a question, what is the zone in Fig. 4?

I advise you to change the structure of Figure 5. Little is clear from this comparison. For example, let the upper half show an experiment and the lower one show the calculation. And render in a density gradient (numerical shlieren).

Where did the experimental data in Table 2 come from? I didn't find them in [11]. In addition, nothing is said at what points these data were obtained and how and with what sensors they were recorded. At least a brief description of the methods and errors. Or write that in detail about all this is written in [...].

line 229: "Predicted pressure coefficient and actual pressure coefficients"?

line 231: "Predicted heat flux and actual heat fluxes"?

line 237: "actual temperature stress are similar with predicted temperature stress"

What is a paragraph of text on lines 232-237 for? There is almost no semantic load, there is a lot of tautology. Analyze these figures. Or, if there is nothing to analyze there, take it all away and leave only Figure 10.

lines 278-279: The incoming flow conditions and material properties of stainless steels (remove s) cylinder are given in the following tables. Same goes for tab.A2 caption.
